# Supporting Meteorologists in Data Analysis through Knowledge-Based Recommendations

Thoralf Reis * 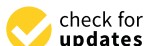, Tim Funke, Sebastian Bruchhaus, Florian Freund, Marco X. Bornschlegl and Matthias L. Hemmje

Faculty of Mathematics and Computer Science, University of Hagen, Universitätsstrasse 1, D-58097 Hagen, Germany
* Correspondence: thoralf.reis@fernuni-hagen.de

**Abstract:** Climate change means coping directly or indirectly with extreme weather conditions for everybody. Therefore, analyzing meteorological data to create precise models is gaining more importance and might become inevitable. Meteorologists have extensive domain knowledge about meteorological data yet lack practical data analysis skills. This paper presents a method to bridge this gap by empowering the data knowledge carriers to analyze the data. The proposed system utilizes symbolic AI, a knowledge base created by experts, and a recommendation expert system to offer suiting data analysis methods or data pre-processing to meteorologists. This paper systematically analyzes the target user group of meteorologists and practical use cases to arrive at a conceptual and technical system design implemented in the CAMeRI prototype. The concepts in this paper are aligned with the AI2VIS4BigData Reference Model and comprise a novel first-order logic knowledge base that represents analysis methods and related pre-processings. The prototype implementation was qualitatively and quantitatively evaluated. This evaluation included recommendation validation for real-world data, a cognitive walkthrough, and measuring computation timings of the different system components.

**Keywords:** information systems; meteorology; big data applications; artificial intelligence; data visualization



## 1. Introduction and Motivation

Climate change impacts people's daily lives in various ways; environmental catastrophes such as flash floods, forest fires [1], or droughts [2] directly deteriorate people's living conditions or indirectly impact people's wealth through food or insurance prices. Meteorological data analysis to provide precise prognoses globally gains importance due to the *"visible increase in potentially irreversible effects of climate change"* [1]. Millions of weather stations and satellites [1] record properties, e.g., wind speed, rainfall, and temperature with high precision. These recordings result in vast amounts of data (volume) that grow with every new measurement (velocity) recorded worldwide, following different national standards (variety) and thus fulfilling the criteria to be classified as Big Data [3].

The advances in the analysis of Big Data in general through algorithms, tools, and the growing support of **Artificial Intelligence (AI)** bear the potential in the meteorological domain *"to mitigate climate change effects"* [1] and even save lives through early warnings [1]. Experts in data science or AI are rare due to a high demand in various industries [4]. This shortage requires non-expert users to analyze data themselves, lacking deep computer science expertise. The authors of this paper introduced with AI2VSI4BigData, a reference model for AI-supported Big Data analysis [4] to overcome these challenges through the application of statistical and symbolic AI. This paper's objective is to introduce a novel recommendation system that supports the meteorological user group in analyzing Big Data. This paper aims to develop a Big Data analysis system that applies symbolic AI to help non-technical users exploit the potential of meteorological Big Data by recommending

suitable data analysis methods. For this purpose, the state of the art is systematically analyzed to select a proper recommendation methodology, and the meteorological user group is assessed towards capabilities and use cases.

The remainder of this paper consists of an overview of the state of the art in AI2VIS4Big Data, AI-based recommendations, and knowledge representations as well as symbolic AI (Section 2), the conceptual model (Section 3), the prototypical implementation (Section 4), and its quantitative and qualitative evaluation (Section 5). The paper concludes by outlining the results and defining future research directions (Section 6).

## 2. State-of-the-Art

This paper aims to develop an **Information System (IS)** for symbolic AI-based recommendations to ease Big Data analysis for non-technical end users. Reference models provide precise terminology and bear the potential to reuse existing implementations. Consequently, state-of-the-art commences with an introduction of the AI2VIS4BigData Reference Model that covers this application domain, followed by an introduction of AI-based recommendations and knowledge representations and symbolic AI.

### 2.1. AI2VIS4BigData Reference Model

The AI2VIS4BigData Reference Model for visual Big Data Analysis defines logical relationships and terminologies and provides guidelines and reference implementations for IS incorporating AI. The relationship between AI and the IS could be using AI for data transformation, empowering its users, or enabling the design and definition of AI models themselves [4]. It was derived by combining Bornschlegl's IVIS4BigData Reference Model with the AI model system life cycle [4]. It provides support for statistical AI, **Machine Learning (ML)**, as well as symbolic AI [4] and aims to fulfill the user journey shown in Figure 1.

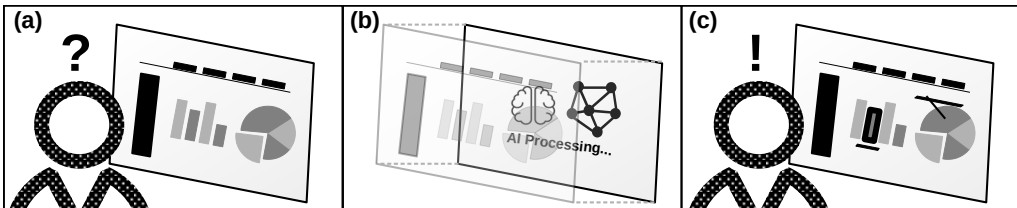

**Figure 1.** AI2VIS4BigData user empowerment user journey [5]. (**a**) Overstrained user; (**b**) AI application; (**c**) communication of insights as user empowerment

The user journey in Figure 1 is based on Fisher and Nakakoji's multi-faceted architecture [6] to support technically less skilled end users in effectively analyzing data [5]. It consists of three steps:

(a) An end user shows the intention of visually analyzing data [5]. The user is thereby not very confident and overstrained by the systems and data complexity [5].

(b) The application of AI on the available information targets to identify areas of relevance or promising next data processing steps [5].

(c) The AI-based result is presented to the user for empowerment [5]. Visual and textual guidance supports the user in effectively using the IS [5].

The authors of this paper detailed this user journey by introducing four user-empowering use cases with 19 potential application scenarios alongside the Big Data analysis pipeline [7]. An example is the use case of AI-based data transformation with the application scenario of predicting analysis methods [7]. As these use cases have been only theoretically introduced, no reference implementation exists. An initial study with nine professionals in the medial application domain positively validated the need for end-user empowerment in visual Big Data analysis [8] and thus strengthened the demand for a reference implementation. Furthermore, no publication details the usage of symbolic AI and the required knowledge representation for the reference model.

### 2.2. Expert and Recommender Systems for AI-Based Recommendations

AI-based recommendations are essential to people's private and professional digital lives. Examples of personal applications comprise product recommendations on e-commerce providers such as Amazon [9], and the advice of suiting movies and shows by streaming providers such as Netflix, that analyze user's preferences and usage data [10]. Examples for professional applications include the selection of potential job candidates for HR [10] and the support of employees through domain-specific recommendations, e.g., medical diagnosis recommendations or mechanical repair instructions [11]. Two possible AI methods for recommendations are **Expert Systems (ES)** [9] and **Recommender Systems (RS)** [9,10] as shown in Figure 2.

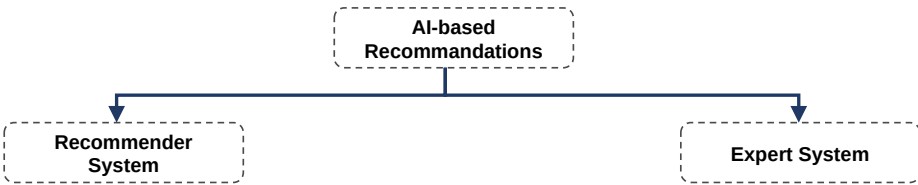

**Figure 2.** Taxonomy of AI-based Recommendations [9].

ES have existed in AI research for approximately fifty years [9,11]. They are a method of AI that incorporates experts' knowledge to solve problems for non-expert users [9]. Problem-solving of ES comprises diagnosis, planning, classification, and others [11]. ES as an application of symbolic AI potentially has the *"capability to review its reasoning and explain its decisions"* [11] and thus is suitable for **explainable AI (XAI)**. Villegas-Ch et al. present an exemplary application of an ES for recommendations with an ES that recommends activities to students to improve their academic performance [12]. Figure 3 shows the five types of ES: *"Rule-based ES, Frame-based ES, Fuzzy-based ES, Neural ES, and Neuro-fuzzy ES"* [9]. They differ regarding their knowledge representation and logic concept. Rule-based ES encode domain knowledge as rules and facts and enable binary reasoning [13]. Frame-based ES use the concept of frames that store knowledge as classes and instances of classes with attributes and values [14]. Fuzzy ES can deal with imprecision and uncertainty instead of Boolean logic [13]. They operate on *"a collection of fuzzy membership functions and rules"* [13]. Neural ES apply the machine learning approach of neural networks for decision-making [13]. Neuro-Fuzzy ES combine the neural and fuzzy approaches.

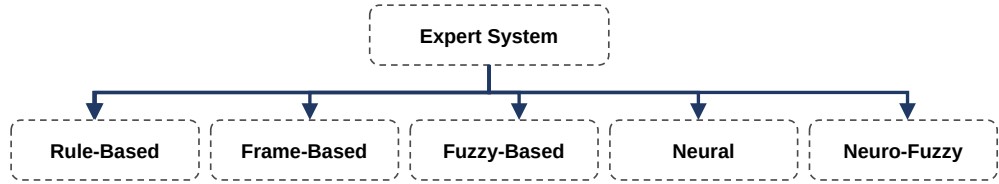

**Figure 3.** Taxonomy of Expert Systems [9].

Figure 4 shows the essential components of an ES: the **Knowledge Base (KB)**, the inference engine, and the user interface. The KB consists of knowledge *"as both facts and heuristics"* [9] that was provided by expert users to the ES. The inference engine processes these facts and heuristics to validate an objective (backward chaining) or to infer new knowledge (forward chaining) [9,12]. Finally, a user interface presents the inference result to the system's end users [9,12].

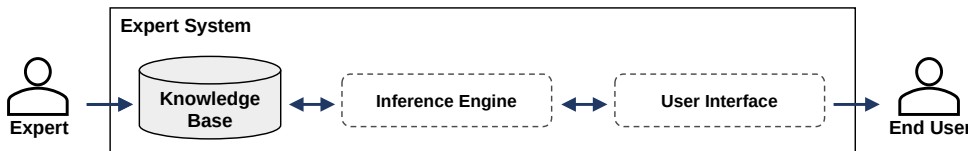

**Figure 4.** Schematic architecture of an Expert System [9,11].

*"Recommender Systems provide suggestions to the users, tailored to their needs"* [9]. These suggestions relate to items, that are *"of use to a user"* [15]. For this purpose, RS internally apply statistical AI and ML [9] on data the end users created. In contrast to this, ES only facilitate the expert users' knowledge [9]. Fewer dependencies on expert users reduces costs and maintenance effort and thus is a major advantage of RS [9]. Another big success factor is the personalized nature of the recommendations. According to Sulikowski et al., personalized recommendations *"lead to high purchase intentions in e-commerce"* [15]. Despite these effects, there also exist non-personalized RS [16]. These non-personalized RS suggest items based on the usage data of all customers [16]. Consequently, all customers receive the same recommendations. Since this does not leverage the big advantages of RS, most applications create user profiles for individual users to provide personalized recommendations. These user profiles are *"based on user demographic data and his/her behavior and transactions"* [17].

Personalized RS can be divided into content-based, collaborative-filtering, demographic, knowledge-based, community-based, and hybrid RS [15]. Content-based RS aim at "presenting products similar to products the user liked in the past" [17]. Collaborative-filtering RS are the most popular RS [17]. They recommend items based on the preferences of other users with similar user profiles [15]. Community-based RS suggest items based on the user's friend's preferences [15]. Knowledge-based RS improve recommendations by applying domain knowledge provided by experts on the items [15]. Demographic RS derive recommendations only based on demographic information within the user profiles [15]. Depending on the granularity of demographic data, this could mitigate the benefits of personalization. Hybrid RS combine two or more recommendation strategies [15]. Figure 5 shows the resulting taxonomy of RS.

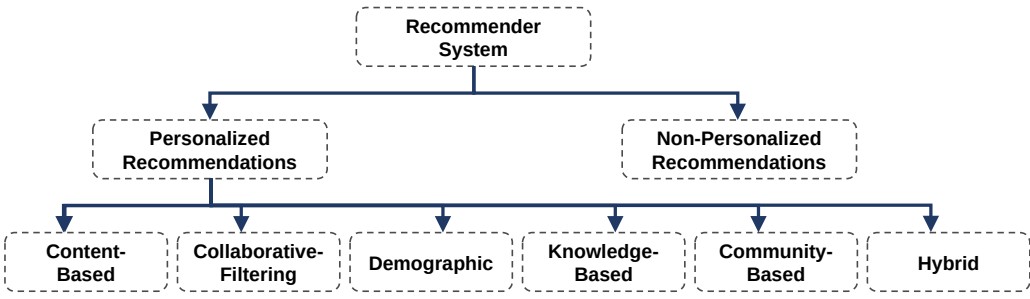

**Figure 5.** Taxonomy of Recommender Systems [15–17].

Although the presented state of the art summarizes many practical applications of ES and RS [10,11,15], it does not explain how an AI-based recommendation system for Big Data analysis in the domain of meteorology shall be designed.

*2.3. Knowledge Representation and Symbolic AI*

Knowledge represents connected, accumulated semantic information [18] that is applied in a task or to solve a problem (also referred to as intelligence in this context) [19]. The aggregated knowledge stored within an IS for expert-based symbolic AI is called KB [14]. AI utilizes a KB to conclude (expert system) [20] or to enhance or train AI models (Statistical AI, Machine Learning). Domain experts perform updates of the KB [14]. At least four different approaches can represent knowledge: representation by using logic, by incorporating semantic networks, by using frames [14], and by defining procedures [19]. The first three approaches collect declarative knowledge, e.g., information about objects, relationships, and backgrounds. They can form a KB, whereas procedural knowledge focuses on directions about particular tasks and solutions for specific problems [19] (implemented, e.g., through an algorithm). An example of declarative knowledge represented by logic is **First-Order Logic (FOL)**. The KB is formed in FOL by a set of statements (or rules) that enables validating hypotheses or drawing conclusions from it [21].

The KB is the centerpiece of symbolic AI; it uses *"logical representations"* [20] within the KB to *"deduce a conclusion from a set of constraints"* [20]. The research focus shifted

from the early introduction of symbolic AI in the 1950s toward statistical AI and ML (with the triumph of a deep convolutional neural network in the ImageNet challenge in 2012 as key milestone [22]). Nevertheless, certain application domains such as planning or optimization still rely on it [20]. Furthermore, the growing importance of XAI leads to a symbolic AI renaissance. Researchers rediscover symbolic AI models due to *"their capability to generate explanations about their processes"* [21] for sensitive domains or to fulfill regulatory requirements. Examples comprise avoiding gender, and ethnic biases in recruitment [10] or enabling audits in financial risk management [21]. Programming languages for symbolic AI comprise, e.g., Prolog, Datalog, and LISP [21], capable of declarative, logical, and symbolic programming paradigms. Although symbolic AI and its knowledge representations have existed for a long time, the state of the art does not describe a KB for Big Data analysis methods.

### 2.4. Discussion and Remaining Challenges

A state-of-the-art review for AI2VIS4BigData, knowledge representations, and symbolic AI revealed four **Remaining Challenges (RC)**: the lack of a reference implementation for AI2VIS4BigData user-empowering use cases [5] (RC1), an AI-based recommendation expert system for meteorologists (RC2); a KB representing data analysis methods (RC3); and a specification of symbolic AI usage for AI2VIS4BigData (RC4). The remainder of this paper addresses them.

## 3. Conceptual Modeling

This paper follows Norman's **User-Centered Design (UCD)** approach. As the name implies, UCD aims to design systems with a special focus on its target users [23]. This focus involves the user groups in the different phases of context definition, requirement derivation, concept design, and evaluation either directly (e.g., via studies) or indirectly by theoretically analyzing their characteristics. Addressing the four remaining challenges from the previous section and following UCD simultaneously, this section begins with analyzing the meteorological use context, deriving requirements, and thus paving the way to solve RC1 in Section 4. Further remaining challenges are answered in Section 3.2 (RC2), Section 3.3 (RC3) and Section 3.4 (RC4).

### 3.1. Meteorological Use Context and Requirements

The core element of the UCD design approach is focusing on the target user group. This paper follows a two-step approach to achieve this for users with meteorological backgrounds: analyzing the user capabilities and researching practical use cases. The results of both steps are then translated into requirements for the development object.

The typical starting point of the professional careers of meteorologists is usually a dedicated study majoring in meteorology. The proposed methodology to identify common minimum computer science skills is analyzing meteorology curricula. As there exists no comparable analysis, the authors of this paper analyzed 17 Bachelor's and Master's meteorology programs in nine German universities. The authors divided the courses in the programs into five categories: meteorology, physics, mathematics, computer science, and chemistry. The results are weighted according to credits, weekly duration, or course occurrence (in this order depending on the available information) and visualized in Figure 6.

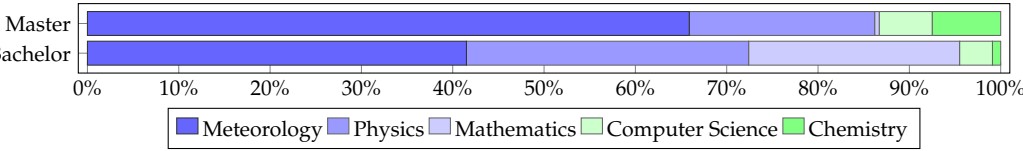

**Figure 6.** Curricular analysis of 17 Bachelor and Master meteorology programs.

The curricular analysis in Figure 6 reveals that the investigated meteorology programs focus strongly on meteorology and physics. In contrast, the share of computer science is minimal (approximately 4% in Bachelor's and 6% in Master's curricula). This finding validates the underlying hypothesis in this paper's introduction. In conclusion, the system needs to bridge the lack of computer science knowledge, including the knowledge about analysis methods and their parameters.

As data drives meteorological research in many ways, the following use cases are only an excerpt. Nevertheless, they enable deriving data analysis requirements of proven practical relevance; scientists and researchers in the meteorological domain describe the application of statistical methods and ML in their publications. These are examples of analysis methods that the aspired system shall recommend.

In [24], Wu and Peng analyze real wind data and weather conditions to forecast the wind power of wind farms [24]. They propose a three-step approach; data pre-processing for data cleaning, normalization, and feature selection; clustering of similar days *"to mitigate the impact of the diversity of training samples"* [24] by applying **K-means Clustering (KM)**; and finally forecasting the wind power [24]. Kovac-Andric et al. utilized meteorological measurements, e.g., sun radiation time, temperature, visibility, and pressure to predict the ozone concentration in eastern Croatia [25]. The authors benefited from the availability of ozone measurements close to the meteorological observation location [25]. The applied **Principal Component Analysis (PCA)** *"to investigate influence of meteorological variables on ozone concentrations"* [25] and **Linear Regression (LR)** *"to estimate how the ozone concentrations depend on meteorological parameters"* [25]. Yamac presents another meteorological data use case in [2]. The author applied AI to meteorological data, e.g., air temperature, solar radiation, wind speed, and humidity for agricultural water management [2]. In more detail, he applied the AI algorithms **k-nearest Neighbor (KNN)**, **Support Vector Machine (SVM)**, and **Random Forest (RF)** to predict crop evapotranspiration, a crucial parameter for optimizing the water stress-sensitive crop of sugar beet [2].

The three practical meteorological data use cases applied the algorithms KM, PCA, LR, KNN, SVM, and RF. Therefore, a system intending to support users with meteorological backgrounds shall support at least these algorithms. Meteorological users lack computer science expertise. Thus, the system shall not only help them in selecting suitable data analysis methods. It shall also inform them about potential pre-processing steps to make data analysis methods applicable. Another requirement is that the system is extensible for different analysis methods and pre-processing. All derived use cases are visualized in Figure 7. The extension of the system with additional analysis and pre-processing methods are two expert user use cases. Meteorologists, as end users, request and consume recommendations (two use cases). The "consume recommendation" use case requires the system to recommend an analysis method, pre-processing, or both (two use cases linked with the include notation).

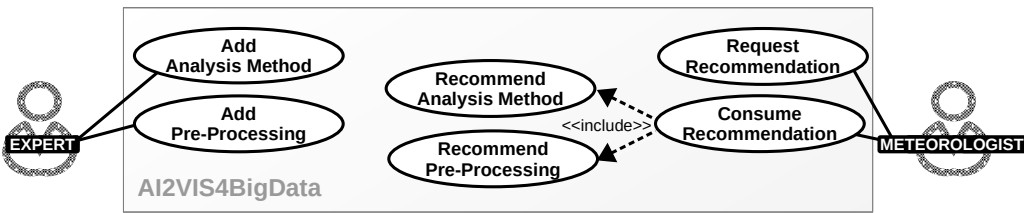

**Figure 7.** Expert and Meteorological End User Use Cases.

### 3.2. Modeling the Recommendation System for Meteorologists

Many forms exist for implementing a recommendation system that enables the use cases introduced in Figure 7 as the multiple methodological alternatives in Section 2.2 reveal. A recommender system with personalized recommendations suits scenarios with a large number of analysis methods and different personal preferences of meteorologists. Yet, it is unsuitable for an IS used by only a few meteorologists. The reason for this is

the challenge of *"limited coverage"* [16] where personalized RS cannot provide suitable recommendations if there is only a little information on the rating of similar users. Non-personalized recommender systems are well suited for smaller user groups and thus a potential choice. Nevertheless, they face two significant challenges in the given scenario. The *"cold-start problem"* [16] complicates the extensibility of new analysis methods. The lack of computer science expertise (revealed by the curricular analysis) raises questions about whether crowd-sourced recommendations can fulfill the objective of only recommending technically suitable analysis methods. An expert system where the expert users must extend a KB upon introducing a new analysis method could overcome these challenges. The knowledge in the KB can follow binary logic as analysis methods are either applicable or not. This paper thus proposes a rule-based recommendation ES that could be extended on a non-personalized RS to prioritize between suitable recommendations. Figure 8 shows the reasoned selection for the conceptual model in the taxonomy.

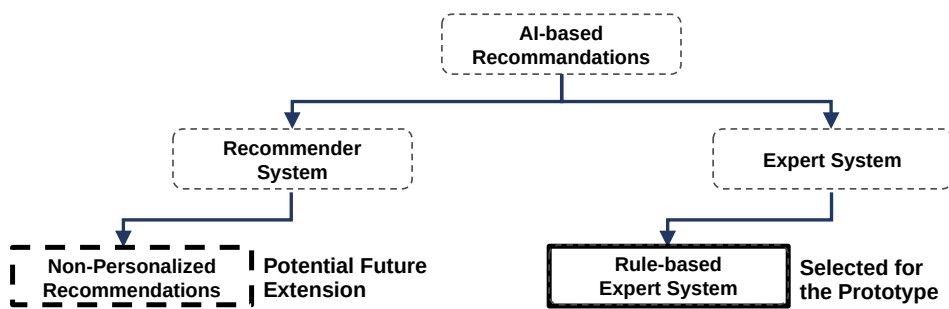

**Figure 8.** Categorization of the selected approach in the recommendation system taxonomy.

Besides the expert users' knowledge base on analysis methods, the proposed expert system's inference engine requires data features. These features can either be automatically determined (e.g., missing values) or need to be manually annotated (e.g., data type categorization as ordinal, numeric, or categorical). A five-step recommendation expert system that suggests analysis methods using a KB is proposed to consider this. The five steps comprise manual rule definition by data science experts (1), manual data annotation by data science experts or meteorological end users (2), automated feature extraction for the dataset (3), automated translation into FOL via mediator (4), and automatic recommendation via inference system (5). Manual step 1 has to be performed once a new analysis method is added to the system. Manual step 2 shall be conducted if the KB contains preconditions for a new dataset that cannot be extracted automatically, e.g., the feature whether an attribute is categorical or numeric. All other features are calculated by a feature extraction workflow (step 3). The data resulting from steps 1 to 3 are then translated into FOL expressions by a mediator component (step 4). Finally, the inference system can process these expressions and infer whether a particular analysis method is applicable, applicable after a pre-processing, or not applicable at all.

Figure 9 shows the proposed **Model-View-Controller (MVC)** [26] architecture to implement this workflow. It consists of the KB, data annotations, features, and recommendations in the system's persistence and three user interfaces for expert and end users. The controller consists of a data preparation **Application Programming Interface (API)**, an asynchronous task queue, and the recommendation expert system. The asynchronous task queue was placed in front of the recommendation expert system so that the system can process queries in parallel and users are not slowed down by waiting for the jobs to finish.

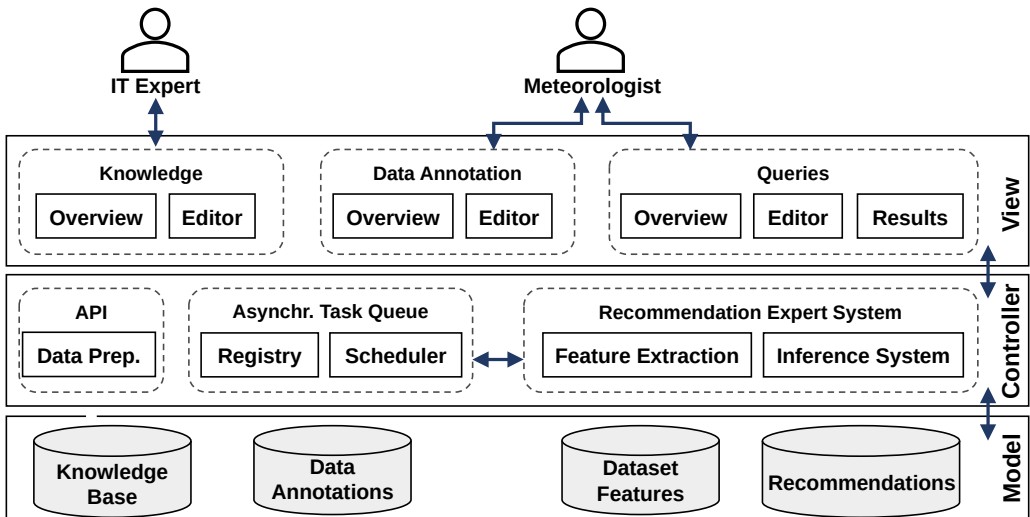

**Figure 9.** Conceptual architecture of the system.

### 3.3. Modeling the Knowledge Base for Data Analysis Methods

Evaluating the applicability of various methods and deriving suggestions to support end users in selecting suitable analysis methods with an expert system uses Symbolic AI. As this requires modeling a KB, the knowledge representation approach must be decided first. This paper chose a declarative KB using FOL since it is well suited for application scenarios that target approving or falsifying a hypothesis.

There exist analysis methods whose applicability can be determined based on formal criteria, whereas others are generally suitable for all problems, while their efficiency strongly depends on data and parameters. The approach outlined in this paper focuses on the former and can be adapted for any simple or complex analysis method. For demonstration purposes, six analysis methods utilized by the use cases in Section 3.1 were selected and are shown in Table 1. Following this selection, seven preconditions determine the applicability of each analysis method. The KB then contains all analysis methods, all preconditions, and their relationship.

**Table 1.** Comparison of analysis methods and preconditions.

| Precondition | KM | PCA | LR | KNN | SVM | RF |
|---|---|---|---|---|---|---|
| supervised | | | X | X | X | X |
| numeric attributes | X | X | X | X | | |
| numeric source attributes | | | | | X | |
| categorical target attributes | | | | X | X | X |
| normalized source attributes | | | | X | X | |
| standardized attributes | X | X | | | | |
| no missing values | X | X | X | X | X | |

The high-level preconditions in Table 1 were then further divided into attribute-related atomic facts to implement a hierarchical rule structure in the KB. The hierarchical structure and pre-processing rules enable the KB to derive suitable analysis methods and suggest necessary pre-processing. Figure 10 shows the breakdown for the precondition "no missing values" from dataset level to attribute level (r1) and further into either no missing values for each attribute (r2) or missing values yet applying proper pre-processing (r3).

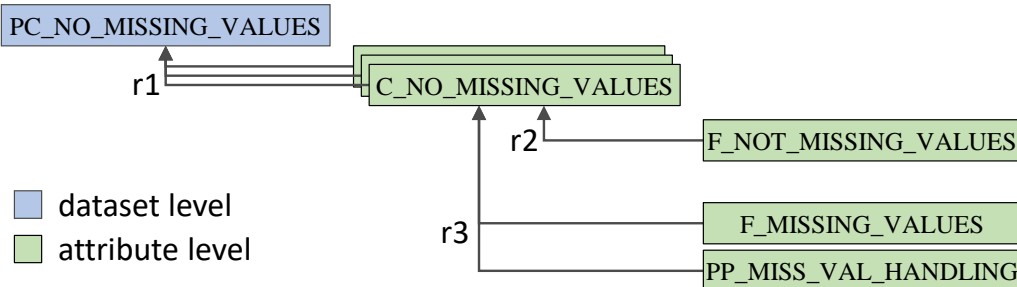

**Figure 10.** Hierarchical structure of the KB for the example of missing values.

### 3.4. Integrating the Concept in AI2VIS4BigData

Figure 11 visualizes the integration of the conceptual ideas and architecture introduced in this paper into the AI2VIS4BigData Reference Model. Recommending analysis and pre-processing methods cover the symbolic AI aspect for the AI2VIS4BigData processing step *Data Analytics* (1). It also fits the reference model's user empowerment concept (2). The extracted features can serve as content guidance and the produced recommendations as interaction guidance (3).

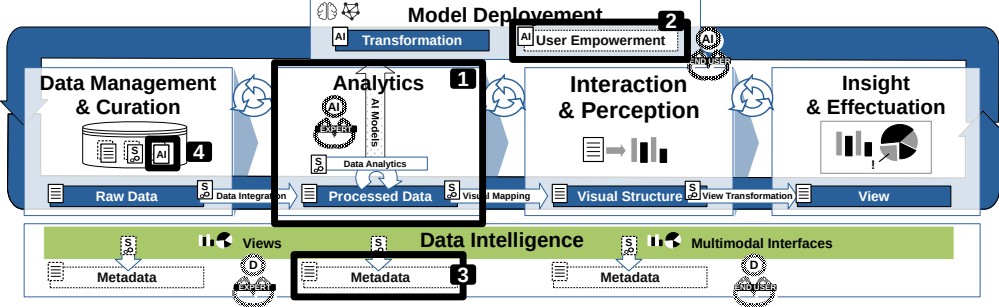

**Figure 11.** Conceptual model integration into AI2VIS4BigData Reference Modell, © 2022 IEEE. Reprinted, with permission, from Reis et al. [8].

A necessary extension of the reference model can be derived from the manual data annotation step. This aspect has not been discussed for AI2VIS4BigData yet is mandatory for features that cannot be created automatically. It fits well with the already incorporated user empowerment concepts from Fisher and Nakakoji where *"domain knowledge should be built into a seed [and] as users use the environment constantly, this seed should be extended"* [6]. Thus, (expert) users need to be able to annotate structured knowledge about the data and provide this information to transformation and user empowerment services as input.

The **User Interface (UI)** concept was introduced by the authors of this paper in [5] and contains interaction as well as content guidance. In this paper's conceptual model, the UI concept is extended on recommendations for suitable analysis methods as interaction guidance, a textual description of applicable analysis methods, and suitable pre-processings as content guidance. Integration into the reference model's UI concept is visualized in Figure 12.

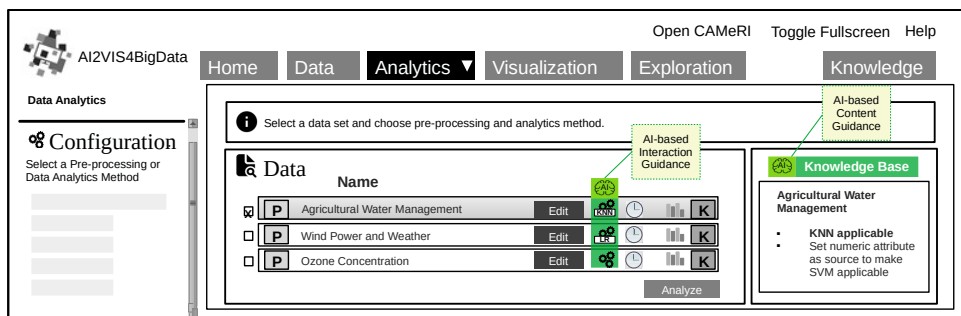

**Figure 12.** Integration of extracted data insights into the AI2VIS4BigData UI concept [5].

## 4. Proof-of-Concept Implementation

The proof-of-concept implementation of the conceptual architecture shown in Figure 9 started with the selection of suiting technology, the design of the technical architecture, and the user interface. The prototype was named **Customizable Analysis Method Recommendation Intelligence (CAMeRI)**. The resulting prototypical implementation can be accessed at the following repository: https://gitlab.com/TimFunke/master-thesis-cameri, (accessed on 8 September 2022).

### 4.1. Technical Architecture

The backend of CAMeRI consists of an API, which was developed with the Python framework FastAPI and the ORM framework SQLAlchemy. Justified by its open-source nature, MariaDB was chosen as the database. The asynchronous task queue was implemented using the Python framework Celery combined with a Redis database as a task queue due to its high performance. Figure 13 shows the described technical architecture of CAMeRI.

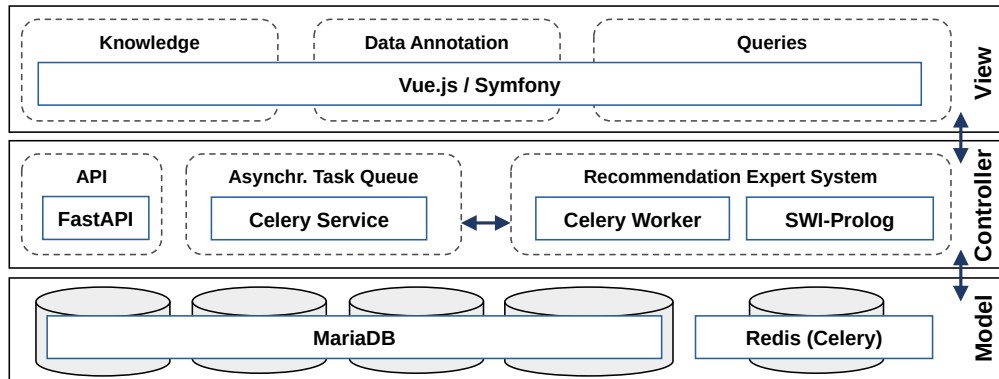

**Figure 13.** Technical architecture of the proof-of-concept implementation.

### 4.2. Knowledge Base Implemetation

The technical architecture applies SWI-Prolog as an inference engine. The declarative KB uses FOL and has a hierarchical structure. It is introduced in Section 3.3. The data features thus need to follow the prolog syntax. To ease understanding, CAMeRI's KB uses prefixes: "am" for analysis method, "pc" for precondition, "c" for columns of the data, "f" for feature, "da" for data annotation, and "pp" for pre-processing. The following listings define the logical hierarchy for a boolean variable "am_pca" that describes the applicability of an analysis method. The syntax is divided into multiple clauses that consist of a head and body separated with the ":-" operator. Each clause is terminated via a "." separator. Components separated by a "," are logically connected via "AND" relationships. Clause heads that appear in multiple clauses on the left side are logically connected via "OR" relationships. The first statement for "am_pca" is shown in Listing 1.

**Listing 1.** KB for analysis method PCA with three preconditions.

```
1  am_pca :-                    pc_no_missing_values ,
2                               pc_numeric_attr ,
3                               pc_standardized_attr .
```

Listing 1 describes for the data analysis method PCA that its application depends on three preconditions: no missing values, numeric attributed, and standardized attributes. The first two preconditions are detailed in Listings 2 and 3.

**Listing 2.** KB for precondition "no missing value".

```
1  pc_no_missing_values :-    c_no_missing_values(col_0),
2                             c_no_missing_values(col_1),
3                             c_no_missing_values(col_3).
4  c_no_missing_values(X) :-  f_not_missing_values(X).
5  c_no_missing_values(X) :-  f_missing_values(X),
6                             pp_miss_val_handling(X).
```

Listing 2 shows the KB entries for a data set with three columns. It demonstrates the hierarchical nature of the KB with the example introduced earlier in Figure 10. The precondition on the data set level is thereby broken down to preconditions on column level ("c") that are either fulfilled by automatically extracted features ("f") or require a "missing value handling" pre-processing ("pp"). The term "c_no_missing_values" is modeled by the expert user in the KB by a placeholder. Subsequently, the mediator component inserts the term for each non-ignored column. In the given example in Listing 2, column 2 is ignored.

**Listing 3.** KB for precondition "numeric attributes".

```
1  pc_numeric_attr :-         c_numeric_attr(col_0),
2                             c_numeric_attr(col_1),
3                             c_numeric_attr(col_3).
4  c_numeric_attr(X) :-       da_type_numeric(X).
5  c_numeric_attr(X) :-       da_not_type_numeric(X),
6                             pp_ignore(X).
```

Listing 3 displays the preconditions of having numeric attributes for three-column data set. For each column, the data is required to either be annotated ("da") as numeric. Alternatively, the respective column needs to be ignored for the application of PCA.

### 4.3. Recommendation Workflow

The workflow for recommending analysis methods was implemented using automatic feature extraction and inference. Figure 14 shows the structure and components of this workflow, which is executed by a task queue (Celery). Initially, the feature extraction automatically calculates features from the corresponding data set and persists them. The controller component of the inference system then loads all the data required for inference, such as features and knowledge base entries. These are then translated by the mediator component into the FOL representation of Prolog and stored in Prolog files. With this, the execution of the Prolog program can be started, whereby the Prolog wrapper component is the interface between Python and SWI-Prolog (https://www.swi-prolog.org, (accessed on 8 September 2022)). The result of the Prolog execution is then read (Prolog Wrapper), translated (Mediator), and finally persisted (Controller).

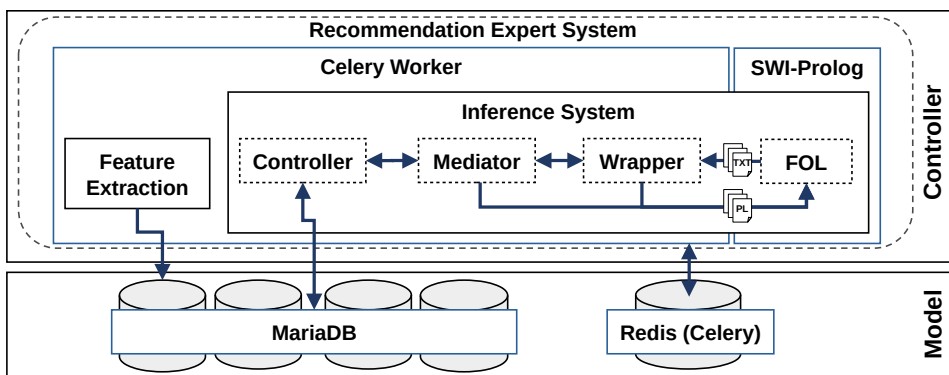

**Figure 14.** Details of the recommendation expert system.

### 4.4. User Interface

The user interface is divided into a query overview, data annotation, and a knowledge center. These are accessible in the header of every page. Under query overview, a user

can create and run queries against the "Recommendation Expert System" and view their results. Under data annotation, data sets already integrated into the system can be extended (annotated) with features that cannot be determined automatically. Finally, the rule-based knowledge base can be managed under the knowledge center. Figure 15 shows the query overview for three data sets. Two queries have been executed in the screenshot, while one query can still be started (play button).

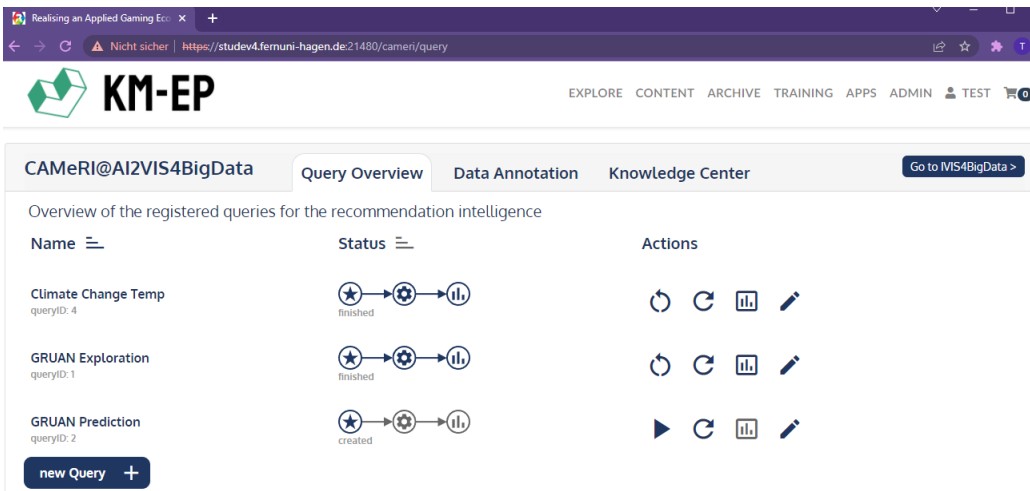

**Figure 15.** Query overview for executed query.

## 5. Evaluation

The evaluation of the CAMeRI prototype implementation addresses qualitative and quantitative aspects. The qualitative assessments review the correctness of the system's analysis method recommendation and the prototype usability. A quantitative evaluation was conducted to examine the system's performance characteristics.

### 5.1. Qualitative Evaluation of Recommendations

The recommendations of the ES are not user-specific, so an evaluation of the proposals, in general, is sufficient. Furthermore, the ES creates its recommendations deterministically by applying the KB, data features, and annotations in its inference system. Hence, a quantitative evaluation of precision, recall, or other **Key Performance Indicators (KPIs)** is not helpful. The system's correct functionality was assessed based on three meteorological test data sets:

(a) Soil Dataset USA 2020 was released on the data science platform kaggle.com (https://www.kaggle.com/cdminix/us-drought-meteorological-data, (accessed on 8 September 2022)) by the author Christoph Minixhofer. It consists of 22 million entries with 21 attributes each. It is compiled from three different sources (NASA LaRC POWER Project, U.S. Drought Monitor, and Harmonized World Soil Database).

(b) City Temperatures World was released on kaggle.com (https://www.kaggle.com/berkeleyearth/climate-change-earth-surface-temperature-data, (accessed on 8 September 2022)) as well. The data set was published by the organization Berkely Earth (http://berkeleyearth.org/, (accessed on 8 September 2022)). It contains 8.2 million measurements of average temperatures in 3448 cities with seven attributes each from 1900 to 2013.

(c) GRUAN Radio Data is a data set authored by the **Copernicus Climate Change Service (C3S)** that was released on their website (https://cds.climate.copernicus.eu/cdsapp#!/dataset/insitu-observations-gruan-reference-network, (accessed on 8 September 2022)). The data was captured by radiosondes during 36,733 flights of meteorological balloons. The measurements were clustered into five bins per flight and averaged for 23 attributes to ease the evaluation.

After annotating the three data sets, the analysis methods from Table 1 were reviewed for correct applicability or pre-processing identification. To ease this assessment, the prototype UI was extended on a result overview (Figure 16).

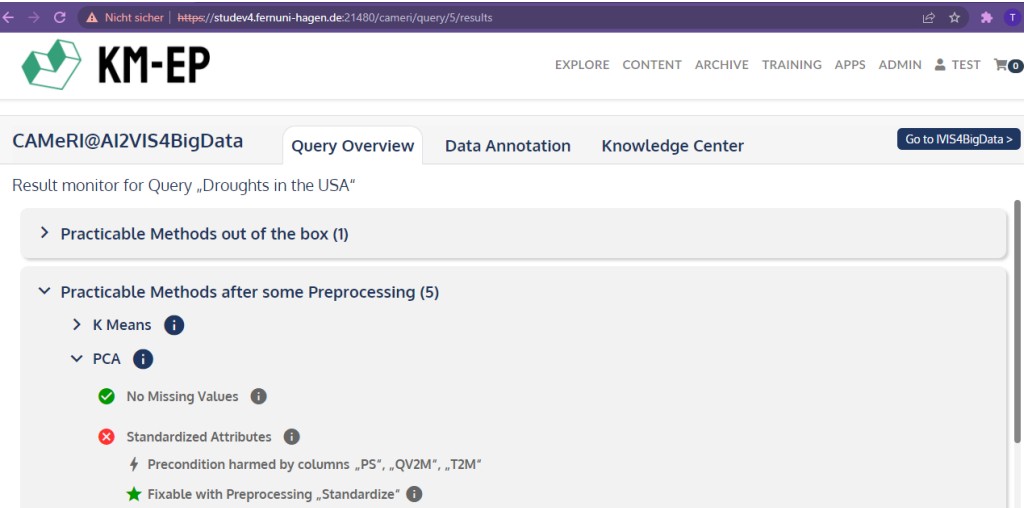

**Figure 16.** Result overview for executed query.

The presented recommendations met the expectation and thus passed the qualitative evaluation successfully. The evaluated analysis methods are grouped on the first level according to their applicability: "Practicable Methods out of the box", "Practicable Methods after some Pre-Processing", and "Impracticable Methods". These groups consist of the actual analysis methods. Below are the verified preconditions, the conflicting attributes, and possible pre-processing methods in the case of a harmed precondition.

## 5.2. Initial Qualitative Evaluation via Cognitive Walkthrough

A **Cognitive Walkthrough (CW)** [27] was applied with a peer group of two experts as an initial qualitative evaluation of the modeled system's usability. The experts comprised an IT manager and a software designer. Due to the prototype development stage, the small peer group aimed to identify significant shortcomings and prepare a more profound assessment with actual end users from the meteorological domain. The following four end-user tasks were selected for the peer group evaluation:

(1)     Complete the data annotation for Soil Dataset USA 2020 data set;
(2)     Create a new query for Soil Dataset USA 2020 data set that includes the whole knowledge base;
(3)     Execute the query from (2);
(4)     Review the applicability of PCA for the query results.

The four tasks were broken down into 21 necessary actions as ground truth before the evaluation. Following the cognitive walkthrough methodology, the evaluators in the peer group had to identify necessary actions for each task by themselves. Figure 17 shows the action of annotating of previously unset columns. As this example shows, the performed actions were single mouse clicks, yet a combination of logically connected steps within one view of the user interface. The example in Figure 17 shows the steps of getting an overview of possible annotations (1) and setting the scale type of one data column (2). Further efforts are making this data column the single target column or not (3) and selecting whether to ignore this data column (4).

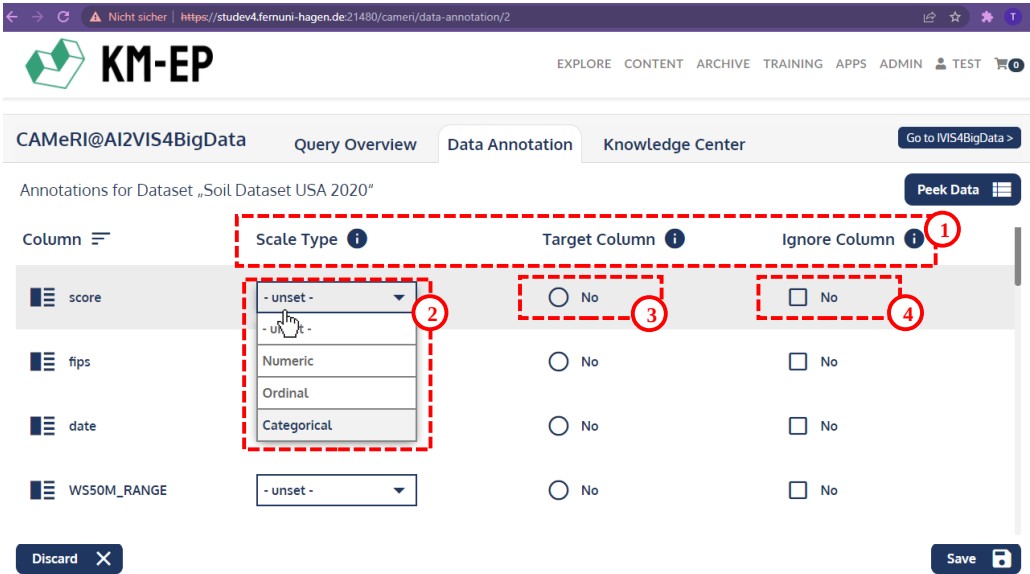

**Figure 17.** Data annotation as example for an action during the cognitive walkthrough.

The selected actions were monitored, recorded, and compared to the ground truth. Furthermore, the evaluators were asked for each task whether they identified any problems, e.g., unexpected effects or unclear actions. The results are presented in Table 2. Table 2 shows that the peer group could successfully identify all specified task actions. A total of 15 (71%) of 21 actions were completed as expected without reporting any problem. Two problems were reported by the IT manager and four by the software designer. This results in a problem rate of only 10% (IT manager), respectively, 19% (software designer). The reported problems primarily addressed the designation of controls for data annotation and missing automation in the user workflow for all screens. Another problem was the lack of automatic status updates for executed queries (mentioned by the software designer). These problems can be traced back to the unfinished nature of the prototypical implementation. They must be addressed before an evaluation with a meteorology end-user peer group can be conducted.

**Table 2.** Cognitive walkthrough results with ground truth actions comparison and identified problems.

| Task | | | | | Evaluator "IT Manager" | | | | Evaluator "Software Designer" | | |
|---|---|---|---|---|---|---|---|---|---|---|---|
| Description | Required Actions | Identified Actions | Problems | Problem Rate | | | | | Identified Actions | Problems | Problem Rate |
| (1) Complete data annotation | 6 | 6 | 2 | 33% | | | | | 6 | 1 | 17% |
| (2) Create a new query | 8 | 8 | 0 | 0% | | | | | 8 | 1 | 13% |
| (3) Execute the query from (2) | 3 | 3 | 0 | 0% | | | | | 3 | 1 | 33% |
| (4) Review the applicability of PCA | 4 | 4 | 0 | 0% | | | | | 4 | 1 | 25% |
| Total | 21 | 21 | 2 | 10% | | | | | 21 | 4 | 19% |

*5.3. Quantitative System Performance Evaluation*

The final evaluation for the current stage of the prototype focuses on assessing the system's performance characteristics. For this purpose, feature extraction and inference time for the three data sets introduced in Section 5.1 were measured. The knowledge base for the measurement contained the necessary knowledge entries for the six analysis methods from Table 1. The measured feature extraction and inference time were derived from timestamps in the Celery queue. Each measurement was repeated five times to avoid coincidental effects. Table 3 shows the resulting average timings for each data set.

**Table 3.** Measurement of duration of feature extraction and inference for three test data sets.

| Test Data | | Measured Duration [s] | |
| --- | --- | --- | --- |
| **Data Set** | **Data Points** | **Feature Extraction** | **Inference** |
| Soil Dataset USA 2020 | 462,000,000 | 30.938 | 0.283 |
| City Temperatures World | 57,400,000 | 5.498 | 0.279 |
| GRUAN Radio Data | 4,224,295 | 0.558 | 0.266 |

The results in Table 3 show rather long durations for feature extraction with a positive linear correlation between feature extraction time and the size of the test data set. Rounded to seven digits after the comma, the relationship between data points per data set and feature extraction time is $1 \times 10^{-7}$. The positive linear correlation is shown by a very strong Pearson correlation coefficient of $r_{FE} = 0.99895$. Inference time is with a value between 200 and 300 milliseconds significantly lower. This result is plausible since the data amount that needs to be processed by the inference engine has been drastically reduced to a fixed number of features per data set attribute. Thus, the potential positive linear correlation of the inference time and the test data set size (Pearson correlation coefficient of $r_I = 0.75585$) is purely coincidental as inference time will correlate with the number of attributes of the data set.

The low inference time in Table 3 raises the question of whether the system would also perform its inference fast enough for a real-time end-user application capable of recommending more analysis methods. Hence, another measurement was conducted to assess the influence of a growing knowledge base. The system's knowledge base was extended from 29 to 250 entries. The second measurement results are shown in Figure 18. The result demonstrates that even for larger KB, the inference time stays reasonably low as it grows for additional 221 KB entries (+862.07%) only for 36 milliseconds (+15.45%).

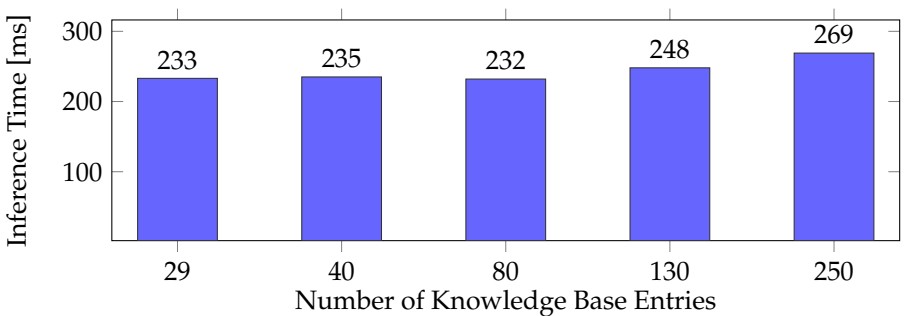

**Figure 18.** Inference time for different knowledge base sizes.

## 6. Summary and Outlook

This paper introduced a recommendation expert system that applies symbolic AI to support meteorologists in analyzing data. In more detail, the paper analyzed the state of the art for recommendation systems, assessed the level of computer science knowledge expected from meteorologists, and derived requirements for the system by reviewing three practical meteorological data use cases to prepare the system design.

The design followed the user-centered design approach. It resulted in a conceptual architecture following the AI2VIS4BigData Reference Model. The concept was implemented as a prototype recommendation system with the name CAMeRI. The concept was then translated into a technical architecture with a detailed knowledge base definition. The resulting software application supports recommendations for six data analysis methods.

The evaluation of the prototype focused on the qualitative and quantitative aspects. Qualitative aspects comprise correct recommendations and usability. Correct recommendations were verified by reviewing all six analysis methods for three real-world meteorology data sets. The system usability was assessed via a cognitive walkthrough with a small peer

group of two evaluators. The quantitative evaluation focused on the assessment of system performance. It was evaluated by measuring the time for feature extraction and inference for the three meteorology data sets and by increasing the knowledge base by introducing further FOL statements.

This paper's contributions are a meteorology curricular analysis, a summary of preconditions for data analysis methods, and a prototype implementation for the recommendation expert system. This system consists of a web-based user interface and backend components for feature extraction, inference, and data handling. The core contribution is an FOL knowledge base specification that supports determining suitable analysis methods and necessary pre-processings.

The remaining challenges include addressing the identified problems during the cognitive walkthrough and repeating the cognitive walkthrough with a larger peer group containing meteorologists. Potential next steps are integrating the created recommendations into the AI2VIS4BigData user interface (Section 3.4) and an extension on a recommender system that prioritizes the guidance of all technically suitable analysis methods and pre-processings based on user preferences.

**Author Contributions:** Conceptualization, formal analysis, investigation, resources, validation, and visualization T.R. and T.F.; methodology, T.R.; software and data curation, T.F.; writing—original draft preparation, T.R.; writing—review and editing, T.R., T.F., S.B., F.F., and M.X.B.; supervision, M.X.B. and M.L.H. All authors have read and agreed to the published version of the manuscript.

**Funding:** This research received no external funding.

**Institutional Review Board Statement:** Not applicable.

**Informed Consent Statement:** Not applicable.

**Data Availability Statement:** The data presented in this paper are available at: https://gitlab.com/TimFunke/master-thesis-cameri, (accessed on 8 September 2022).

**Conflicts of Interest:** The authors declare no conflict of interest.

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
