# Peer review of "Supporting Meteorologists in Data Analysis through Knowledge-Based Recommendations"

_2504-2289, doi:10.3390/bdcc6040103_

Round 1

Reviewer 1 Report (Previous Reviewer 2)

This version took most of my suggestions into account and it is now a paper that can be published. It lacks some novelty for the general public, but it is remarkably interesting for the meteorology sector.

Author Response

Dear Sir or Madam,

Thank you very much for your initial and follow-up review. We are grateful for the provided suggestions. As you require no change after your second review, we leave the content unchanged.

Reviewer 2 Report (Previous Reviewer 3)

The latest manuscript with the revisions clears all the doubts previously discussed. The paper has been substantially improved and now warrants publication, in my view.

Author Response

Dear Sir or Madam,

Thank you very much for your initial and follow-up review. We are grateful for the provided suggestions. As you require no change after your second review, we leave the content unchanged.

This manuscript is a resubmission of an earlier submission. The following is a list of the peer review reports and author responses from that submission.

Round 1

Reviewer 1 Report

This paper presents a recommendation system based on symbolic Artificial Intelligence that facilitate the data analysis to meteorologists recommending suitable data analysis methods.

C1) One of the major drawbacks of this paper is its limited novelty. Mainly, the article focuses on the high-level description of the implemented conceptual model and prototype. A higher scientific soundness was expected. For example, the article lacks of basics related to Recommendation Systems. In order to facilitate the reader’s understanding, it would be good to answer questions such as: Where in the hierarchy of recommender systems is the recommendation approach proposed by the author in this article? Which is the relationship between Users X Items -> Ratings for this scenario? Another example is that 47% of the bibliographic references are out of date (8 of 17 articles are in the year range 1992 to 2015).

C2) Another weakness is the poor evaluation carried out by applying a Cognitive Walkthrough with only two experts (an IT expert and a software designer). Additionally, the article lacks experimental evaluations that validate the quality of the recommendation algorithm used in the proposed information system.

C3) Regarding the implemented prototype, the authors do not indicate where it can be accessed (e.g. GitHub). In fact, the link (https://studev4.fernuni-hagen.de:21480/cameri/query/5/results) shown in Figure 9 is not available.

Reviewer 2 Report

The article is too vague in the parts about the experimentation and results.

According to the abstract, the paper “describes the systematic design approach following a user-centered design approach, the prototypical system implementation, and the evaluation with a cognitive walk-through”. I think that the design part is a bit out of scope of this journal unless it somehow shows some novelty in terms of an intelligent placement of information on screen. In section 4.3, where the authors could demonstrate that, there is only a brief description of the layout with nothing that could be seen as new.

In sections 5 and 6 there is nothing worthy (3 + 2 paragraphs), authors should have made more experimentation and discussion. There is no information on how CW was performed, the results show that there are still some work do be done, and then authors only present a final section “Summary and Outlook”. If one wants to replicate the results, they have no way of doing it.

In terms of readability, there are some sentences too long, some of which are difficult to understand. Some short sentences are also confusing. For instance, the sentences in lines 25-27 “Experts in data science or AI are rare due to a high cross-industry demand [4] so that many meteorologists need to analyze data themselves lacking deep computer science expertise.” and 149-151 “Another requirement is the extensibility of the system for additional methods; the reason for this is that the shown example use cases are only an excerpt. All derived use cases are visualized in figure 3.”. The former has probably an extra “that” but the latter I still cannot understand what the authors meant. More than typos or misspellings the main problem is the construction of sentences that makes the text hard to follow. A revision by a native English speaker would solve this problem.

Reviewer 3 Report

The reviewed paper proposes a solution to help meteorologists who lack significant data analysis skills create better models. The system proposes data analysis methods or data pre-processing to meteorologists using symbolic AI, a knowledge base created by experts and a recommender system. The system is based on the AI2VIS4BigData Reference Model. The authors’ prototype was called CAMeRI (Customizable Analysis Method Recommendation Intelligence).

It is important to appreciate the fact that the authors followed a user-centered design approach, as well as the field of application, whose importance is growing due to climate change.

The paper is quite well organized and easy to follow, with some minor request for elaboration, corrections and references, as stipulated below. 

Please focus on the Abstract to make it more adequate a technical journal, concentrating on the merits of your research, in particular precising the final results of the proposal. 

In Chapter 2. State of the Art I appreciate the provided background for 3 areas but I find that an introduction to recommender systems is lacking. I would appreciate a summary of different recommendation approaches (please follow recommender systems review, chapter 2.1, in paper titled Modeling online user product interest for recommender systems and ergonomics studies) and of main recommendation techniques—advantages and disadvantages (please follow chapter 1 in paper titled Gaze and event tracking for evaluation of recommendation-driven purchase).

Figure 1 is difficult to follow with regard to the text. Please could you try to make it bigger and more legible, and also all notions used in the Figure could be described in the text in a descriptive form?

In Chapter 3 Figure 3 is difficult to follow with regard to the text. Please try to make it bigger and more legible and all notions used in the Figure should be described in the text in a descriptive form – moreover, you have not referred to number 4 in the Figure in a way you did for numbers 1-3. Please correct that.

I would appreciate more elaboration in Chapter 5. It is too brief in comparison to the in-depth content of the remaining parts of the paper. I also wonder if it were possible to prove the applicability thanks to initial tests with selected meteorologists or at least provide information on the authors’ plans regarding that. Please include a wider summary of the main results, as well as information on their limitations and develop future research directions.

I am available to re-review the paper after improvements.